# GENETIC INTELLIGENCE AND TENSOR-UNITARY TRANSFORMATIONS

**S.V.Petoukhov** *

### ABSTRACT

The article is devoted to the author's results of the analysis of the genetically inherited ability of living bodies to intellectual activity (for example, the ability to echolocate in dolphins), which resulted in the emergence of algebraic formalisms called tensor-unitary transformations. Genetic intelligence is understood as that part of the intellectual potential of living organisms that allows, on the basis of genetic information in DNA and RNA molecules, to build, for example, from one fertilized cell an organism with trillions of cells so that the parental traits are reproduced in it in a multichannel noise-resistant manner, despite strong noise and constantly changing conditions of nutrition and external influences during life. In this case, we are talking about the systematic growth in the course of ontogenesis of the number of parameters and degrees of freedom of the body with a corresponding increase in the dimensionality of its configuration space of states. With such growth, the organism at successive stages of its development, acquiring new degrees of freedom and knowledge, somehow retains the memory of the skills and knowledge that it possessed at previous stages of life. The author develops the algebraic foundations for modeling this fundamental feature of the development of living bodies in the tensor-matrix language of systems of multidimensional vector configuration spaces. Tensor-unitary transformations are operators that preserve the lengths of vectors during their tensor transformation into vectors of a space of increased dimension (in contrast to conventional unitary transformations that transform vectors into a space of the same dimension). They are operators of expansion of stochastic-deterministic memory with preservation of all previous memory. Possible applications of tensor-unitary transformations for the development of AI, genetic algorithms, etc. are discussed.

## 1  INTRODUCTION

This article is devoted to the analysis of genetically inherited ability of living bodies to perform intellectual activities (for example, the ability to echolocate in dolphins), as well as algebraic formalisms called tensor-unitary transformations. These transformations are proposed by the author as a result of the said analysis for modeling biological phenomena and possible development of approaches to artificial intelligence systems and new genetic algorithms. In artificial intelligence systems, their creators strive to reproduce the properties of the natural intelligence of living bodies, the structures and mechanisms of which are inherited from generation to generation due to their connection with the genetic coding system. The author studies the structural features of the genetic coding system for the disclosure of information patents of living nature to ensure the inherited abilities of living organisms to perform intellectual activities. Genetic intelligence is understood as that part of the intellectual potential of living organisms that allows, on the basis of genetic information in DNA and RNA molecules, to build, for example, from one fertilized cell an organism with trillions of cells in such a way that the parental characteristics are reproduced in it in a multichannel, noise-resistant manner, despite strong noise and constantly changing conditions of nutrition and external influences during life. In this case, we are talking about the systematic growth in the course of ontogenesis of the number of parameters and degrees of freedom of the body with a corresponding increase in the dimensionality of its configuration space of states. At the same time, with such growth, the organism at the next stages of its development, acquiring new degrees of freedom and knowledge, retains the

---

*Mechanical Engineering Research Institute, Russian Academy of Sciences, Moscow, Russia.
spetoukhov@gmail.com , http://petoukhov.com/

memory of the skills and knowledge that it possessed at previous stages of life. The author develops the algebraic foundations for modeling this fundamental feature of the development of living bodies in the tensor-matrix language of systems of multidimensional vector configuration spaces. The desirable modeling tools should allow modeling the phenomenon of preserving the memory of past states in a developing vector system during its transition to new configuration spaces of increased dimension with the possibility of simultaneously modeling the expansion of memory with a corresponding increase in degrees of freedom. In mathematical natural science, the tensor product of matrices is used to increase the dimensionality of the model vector space. The genetic coding system, unique in its speed properties, noise immunity and general biological significance, together with the family of DNA alphabets, is structured precisely for the tensor product of matrices, which links, for example, matrix representations of different DNA alphabets into a single tensor family of alphabetic matrices (Petoukhov (2008),Petoukhov & He (2023)). The founder of quantum information science, Yu. I. Manin, introduced the concept of a quantum computer in his book Manin (1980) precisely when analyzing the features of high-speed processing of DNA information in chromosomes by "genetic automata". He prophetically pointed out the important role of unitary rotations and tensor products: "A quantum automaton must be abstract: its mathematical model must use only the most general quantum principles, without prejudging physical implementations. Then the evolution model is a unitary rotation in a finite-dimensional Hilbert space, and the model of virtual division into subsystems corresponds to the decomposition of space into a tensor product. Somewhere in this picture there must be a place for the interaction traditionally described by Hermitian operators and probabilities" (Manin (1980)). Thus, the very birth of quantum information science, so promising for the problems of artificial intelligence, occurred thanks to the desire to understand the features of genetic information science. Obviously, the universal rules of probabilities in genomic DNA as multilayered texts written in the languages of tensorially interconnected alphabets of n-plets of DNA, presented in this article, together with the unitary rotations tied to these rules, are consistent with this prediction of Yu. I. Manin (one example of the speed of genetic processes that amazed him is the process of replication of DNA strands in the bacterium E. coli at a speed of more than 1000 nucleotides per second (Bank (2022)).

## 2  FEATURES OF TENSOR-UNITARY TRANSFORMATIONS

The concept and apparatus of tensor-unitary transformations arose in the author as a result of studying the biological dualism of "stochastics-determinism", primarily in the information sequences of genomic DNAs of higher and lower organisms. Individual molecules interact in cells stochastically. In living bodies, everything is associated with stochastics. Even genetically identical cells of the same tissue have different levels of protein expression, sizes, etc. But with this stochastics in the "small", macro-corporeal traits are inherited from parents as determined. All genetics as a science began with Mendel's discovery of statistical rules for the inheritance of traits when crossing organisms. According to Mendel's law of independent inheritance of traits, information from the level of DNA molecules dictates the macrostructures of living bodies through many independent channels, despite strong noises. Thus, the colors of hair, eyes and skin are inherited independently of each other. Accordingly, each organism is a machine of multi-channel noise-resistant coding based on the dualism of stochastics-determinism.

The results presented in this article are based on the universal rules of statistical organization of information sequences of single-stranded genomic DNAs of higher and lower organisms, which the author discovered and which are related to the stochastic-deterministic dualism (Petoukhov (2021),Petoukhov (2022b)). It is logical to look for opportunities to model these universal rules based on the formalisms of quantum mechanics and quantum information science, since DNA belongs to the microworld of quantum mechanics, which is based on the concept of probabilities. In quantum mechanics and quantum information science, unitary transformations play an important role: the evolution of closed quantum systems is described by unitary transformations, and all calculations in quantum information science are based on unitary operators that play the role of logical gates (quantum gates) (Nielsen & Chuang (2010)). Unitary transformations of vectors preserve their lengths and the values of scalar products (in the case of real components, they are orthogonal transformations).

By definition, tensor-unitary transformations are transformations that preserve the norms (lengths) of vectors during their tensor transformation into vectors of a space of increased dimension. Unlike

unitary transformations, which transform the original n-dimensional vector into an n-dimensional vector of the same dimension, tensor-unitary transformations transform the original (or "parent") n-dimensional vector into a "daughter" m-dimensional vector of increased dimension (m>n) while preserving the length of the parent vector. We can say that they ensure the "inheritance" of the vector length into the length of a new tensor-transformed vector belonging to a space of tensor-increased dimension; this new space can be interpreted as the configuration space of a multiparameter system, the number of parameters of which has increased in the course of its development (by analogy with biosystems in the process of their ontogenetic growth).

Applying these tensor-unitary transformations to vectors can be thought of as a two-step process. First, the original n-dimensional vector, called the parent vector, is represented as the sum of all its basis vectors with their coordinate weights (e.g., the vector [x, y] is represented as the sum: x[1, 0]+y[0, 1]). Secondly, each of these weight basis vectors is tensor-multiplied by a so-called "norm-vector" (a qubit-like vector) that ensures that the length of the parent vector is preserved in the tensor-increased child vector. By definition, a norm-vector is a k-dimensional vector $[\alpha_0, \alpha_1, \alpha_2, \ldots, \alpha_{k-1}]$, the sum of the squares of the coordinates of which is equal to 1: $\alpha_0{}^2 + \alpha_1{}^2 + \alpha_2{}^2 + \ldots + \alpha_{k-1}{}^2 = 1$. Its coordinates can be interpreted as probability amplitudes (as in qubit or poly-qubit vectors), which in the general case can be either fixed numbers or functions of time or other variables. We briefly note that tensor-unitary transformations can be represented as a special case of Hadamard products for block vectors, in which we are talking about component-wise multiplication of a vector by the corresponding number of norm-vectors; a special symbol has been proposed to denote tensor-unitary transformations: (its Unicode 0x235d, https://unicodemap.org/details/0x235D/index.html) (Petoukhov (2022a)).

In the case of a tensor-unitary transformation of the parent vector, the "qubit-likeness" of norm-vectors ensures the introduction of an element of probability (stochasticity) into the values of individual coordinates of the daughter multidimensional vector, special groupings of coordinates of which turn out to be deterministic carriers of the exact memory about the coordinates of the parent vector. Thus, tensor-unitary transformations, being stochastically-deterministic transformations, generate stochastically-deterministic vectors and allow modeling the biological dualism of "stochastics-determinism". Let us explain this with a simple example.

Let us consider the simplest 2-dimensional vector function [x(t), y(t)] = [x(t), 0] + [0, y(t)], and tensor multiply its first weight basis vector [x(t), 0] by the norm-vector $[\alpha_0, \alpha_1]$, and its second weight basis vector [0, y(t)] by the norm-vector $[\beta_0, \beta_1]$ (here $\alpha_0{}^2 + \alpha_1{}^2 = 1$, and $\beta_0{}^2 + \beta_1{}^2 = 1$). As a result, we obtain a 4-dimensional vector-function $\overrightarrow{\boldsymbol{D}}_1$:

$$\overrightarrow{\boldsymbol{D}}_1 = [x(t), 0]\otimes[\alpha_0, \alpha_1] + [0, y(t)]\otimes[\beta_0, \beta_1] = [x(t)\alpha_0, x(t)\alpha_1, y(t)\beta_0, y(t)\beta_1] \qquad (1)$$

The values of the individual coordinates of the daughter vector-function $\overrightarrow{\boldsymbol{D}}_1$ (1) are stochastic, since each of them contains one of the probability amplitudes $\alpha_0$, $\alpha_1$, $\beta_0$, $\beta_1$. There are infinitely many possible values of $\alpha_0$, $\alpha_1$, $\beta_0$, $\beta_1$, and their random selection changes the values of these individual coordinates, but the lengths of the child and parent vectors will be equal for all these selections and for each fixed value of the variable t. In other words, different sets of values $\alpha_0$, $\alpha_1$, $\beta_0$, $\beta_1$. give different daughter vector functions $\overrightarrow{\boldsymbol{D}}_1$, which differ from each other in their coordinates, but they all have the same length for any fixed value of the parameter t. This is confirmed by calculating the length $||\overrightarrow{\boldsymbol{D}}_1||$ of the vector-function $\overrightarrow{\boldsymbol{D}}_1$ (1) under any fixed value of the parameter t:

$$||\overrightarrow{\boldsymbol{D}}_1|| = \{x^2\alpha_0^2 + x^2\alpha_1^2 + y^2\beta_0^2 + y^2\beta_1^2\}^{0.5} = \{x^2(\alpha_0^2 + \alpha_1^2) + y^2(\beta_0^2 + \beta_1^2)\}^{0.5} = (x(t)^2 + y(t)^2)^{0.5} \qquad (2)$$

This length (2) of the daughter 4-dimensional vector function $\overrightarrow{\boldsymbol{D}}_1$ is equal to the length of the mother 2-dimensional vector function [x(t), y(t)] at any fixed time t. In other words, the daughter vector function $\overrightarrow{\boldsymbol{D}}_1$, having stochastic coordinates and increased dimensionality, retains an exact "memory" of the length of the mother vector function at any fixed time, regardless of the stochasticity of its individual coordinates, and in this respect is a deterministic entity. We can say that stochastics with hidden determinism takes place here. Now let us note other deterministic properties of this daughter 4-dimensional vector function $\overrightarrow{\boldsymbol{D}}_1$, associated with its projections onto coordinate planes.

A four-dimensional vector space with a Cartesian system of numbered coordinates $[X_0, X_1, X_2, X_3]$ contains 6 coordinate planes:

$$(X_0, X_1), (X_2, X_3), (X_0, X_2), (X_0, X_3), (X_1, X_2), (X_1, X_3) \qquad (3)$$

In the plane $(X_0, X_1)$ the daughter vector function $\overrightarrow{D}_1$ (1) is represented by its projection $\overrightarrow{M}_{01}$ = $[x(t)\alpha_0, x(t)\alpha_1, 0, 0]$, the length of which $||\overrightarrow{M}_{01}||$ (4) is equal to the first coordinate of the parent vector function $[x(t), y(t)]$ for any fixed t:

$$||\overrightarrow{M}_{01}|| = \{x(t)^2\alpha_0^2 + x(t)^2\alpha_1^2\}^{0.5} = \{x(t)^2(\alpha_0^2 + \alpha_1^2))\} = x(t) \qquad (4)$$

In the plane $(X_2, X_3)$ the daughter vector function $\overrightarrow{D}_1$ is represented by its projection $\overrightarrow{M}_{23}$ = $[0, 0, y(t)\beta_0, y(t)\beta_1]$, the length of which $||\overrightarrow{M}_{23}||$ (5) is equal to the second coordinate of the mother vector function $[x(t), y(t)]$ for any fixed t:

$$||\overrightarrow{M}_{23}|| = \{y(t)^2\beta_0^2 + y(t)^2\beta_1^2\}^{0.5} = \{y(t)^2(\beta_0^2 + \beta_1^2))\} = y(t) \qquad (5)$$

Expressions (2, 4, 5) indicate that the daughter 4-dimensional vector-function $\overrightarrow{D}_1$, having stochastic coordinates, contains in the groupings of these coordinates the exact "memory" (or values) of all coordinates and the length of the parent 2-dimensional vector-function, i.e. in this respect it is a deterministic entity. Accordingly, tensor-unitary transformations are in this respect **the operators of stochastic-deterministic memory**

The simplest example considered shows that tensor-unitary transformations and the families of daughter vectors (or vector-functions) generated by them allow modeling the biological dualism of "stochastics-determinism", as well as the phenomena of Gestalt biology, in which only the aggregate values in groupings of elements are significant in contrast to the insignificance of individual elements in themselves, by analogy with the well-known genetically inherited phenomena of Gestalt psychology. For example, we recognize a musical melody even when it is performed on different instruments and in different frequency ranges with a change in the absolute values of the sound frequencies of individual notes, which turn out to be insignificant in contrast to the ratios in groupings of note frequencies. Since tensor-unitary transformations allow modeling biological Gestalt phenomena, they can also be conventionally called Gestalt transformations, and the stochastic-deterministic vectors (or vector functions) generated by them can be called Gestalt-vectors.

It should be emphasized that usually unitary (or orthogonal) operators are understood as unitary square matrices, but in the case of tensor-unitary transformations, the component-wise tensor product of the original vector by the corresponding number of norm-vectors is used to obtain a stochastic-deterministic object. In the general case, the number of norm-vectors is equal to the dimension of the parent vector (or the daughter vector of the previous generation), each component of which must be multiplied by a separate norm-vector.

Note that the daughter 4-dimensional vector $\overrightarrow{D}_1$ (1) is not only a carrier of the exact memory of the coordinates of the parent 2-dimensional vector, but also a carrier of new information in the planes $(X_0, X_2), (X_0, X_3), (X_1, X_2), (X_1, X_3)$ of the 4-dimensional space. In these planes, the vector $\overrightarrow{D}_1$ is represented by its projections $\overrightarrow{M}_{02}$ = $[x(t)\alpha_0, 0, y(t)\beta_0, 0]$, $\overrightarrow{M}_{03}$ = $[x(t)\alpha_0, 0, 0, y(t)\beta_1]$, $\overrightarrow{M}_{12}$ = $[0, x(t)\alpha_1, y(t)\beta_0, 0]$, $\overrightarrow{M}_{13}$ = $[0, x(t)\alpha_1, 0, y(t)\beta_1]$, the lengths of which are not equal to the coordinates of the maternal vector.

The values of their lengths carry new information, which is introduced into them by the stochastic coordinates of the norm-vectors $\alpha_0, \alpha_1, \beta_0, \beta_1$. These new stochastic components can reflect, for example, the specificity of the ontogenesis stage or the impact of the external environment on the organism. Thus, the tensor-unitary transformation is an **operation of memory expansion** (with the preservation of all previous memory) for a tensor-extended multiparameter system, which, along with the memory of the parameters of the "ancestors", contains many new parameters or information. Accordingly, tensor-unitary transformations can be used to model growing stochastic-deterministic

systems of morphogenetic, biorhythmic and other types, where the number of parameters and degrees of freedom increases step by step.

The values of the scalar products of two parent vectors and their two daughter vectors are equal only when the tensor-unitary transformations of both parent vectors use the same set of norm-vectors.

Tensor-unitary transformations can be repeatedly applied to develop from a parent vector increasingly complex multiparameter child vectors with a step-by-step increase in the dimensionality of their configuration spaces. In certain groupings of coordinates of these growing child vectors (i.e. in projections onto subspaces of the configuration space), information about all coordinates of both parent vectors and previous child vectors will be preserved, and new information will also be presented (in projections onto other subspaces). Tensor-unitary transformations allow one to record the step-by-step history of the development of a tensor-growing multiparameter stochastic-deterministic system in the form of sequences of daughter vectors corresponding to the original parent vector and the norm-vectors used.

Let us repeat that the coordinates of norm vectors can be not only fixed values of probability amplitudes, but also normalized functions of time or other variables. Expression 6 shows an example of two-dimensional norm-vectors $[\alpha_0, \alpha_1]$, the coordinates of which are functions of the variable t, and the sum of the squares of the coordinates is equal to 1:

$$||\overrightarrow{\boldsymbol{F}}|| = [\alpha_0(t)/\{\alpha_0(t)^2 + \alpha_1(t)^2\}^{0.5}, \alpha_1(t)/\{\alpha_0(t)^2 + \alpha_1(t)^2\}^{0.5}] \qquad (6)$$

At any value $t$, the sum of squares of its coordinates is equal to 1, that is, the vector function $\overrightarrow{\boldsymbol{F(t)}}$ satisfies the definition of normvectors and can be used in tensor-unitary transformations. The maternal vector can also be normalized, that is, have coordinates whose sum of squares is equal to 1. An example of such a vector:

$$[x(t)/\{x(t)^2 + y(t)^2\}^{0.5}, \ \ y(t)/\{x(t)^2 + y(t)^2\}^{0.5}] \qquad (7)$$

The tensor-unitary product of such a normalized parent vector by a norm-vector yields a vector, the sum of the squares of the coordinates is again equal to one, i.e. it again generates a norm-vector. This allows one to model **multiple reproductions** of parametrically defined geometric configurations using tensor-unitary transformations

For modeling bio-cyclic phenomena, the case when the functions serving as coordinates of the normalized parent vector of type (7) are cyclic, for example, consist of superpositions of sines and cosines, is of particular interest. Then the daughter vectors generated by tensor-unitary transformations will also be endowed with coordinates of a cyclic nature for modeling multiparameter systems consisting of many subsystems of cyclic behavior. The importance of this case is due to the fact that the organism is a huge chorus of coordinated cyclic processes, the number of which increases as it develops ontogenetically from the embryonic to the mature state, with step by step obtaining new and new degrees of freedom, coordinated with the already existing ones. For example, in the organism of an adult, their number reaches enormous values, since it contains approximately 100 trillion cells participating in these sets of coordinated cycles. Moreover, all these coordinated cyclic processes occur against the background of random (stochastic) interactions between individual molecules in cells and are themselves stochastically determined to a certain extent. According to the provisions of ancient chrono-medicine, all our diseases are the result of a violation of this coordination.

Note that tensor-unitary transformations can be applied not only in the case of real numbers, but also in the cases of various systems of multidimensional numbers: complex numbers, hyperbolic numbers, Hamilton quaternions, Cockle split-quaternions, and so on.

So far we have been talking about tensor-unitary modeling of growing multiparametric systems (such as multicellular biosystems in ontogenesis). But we can also talk about similar tensor-unitary modeling of such families of computer memory cells that are gradually expanded in the course of their model development. This makes it easier to understand that tensor-unitary transformations can lead to computer artificial intelligence of the genomorphic type: such intelligence can be based on generations of tensor-unitary expanding families of computer memory cells, in which some groups of cells carry information about the states of all cells of families of previous generations, and other

groups of cells carry new information related to the current effects of the external and internal environment.

For this reason, tensor-unitary transformations and the families of generations of stochastic-deterministic vectors generated by them seem to be promising algebraic tools for creating genomorphic-type artificial intelligence systems. They are also useful for developing genetic algorithms that use the principle of natural selection to "grow" artificial intelligence and are used worldwide on the basis of hundreds of patents for many technical problems Buontempo (2019). The listed algebraic properties of tensor-unitary transformations and the families of generations of daughter multidimensional vector functions generated by them partially clarify why the universal rules of stochastic organization of genomic DNA are structured by nature in accordance with these algebraic operations and families.

In quantum mechanics, unitary transformations describe the evolution of closed quantum systems. The author believes that tensor-unitary transformations are useful for developing the quantum mechanics of open quantum systems, growing with the increase of their multicomponent composition similar to biological bodies developing during ontogenesis and phylogenesis. They are also associated with the tensor-matrix theory of digital antenna arrays and the doctrine of bio-antenna arrays as the basis of energy-information biological evolution (Petoukhov (2022a)). A few additional words should be said here. In our time, special attention is paid to the possibilities of using in evolutionary biology the achievements of mathematical natural science and informatics, for example, from the fields of noise-immune coding and information transmission, physical fields theory, holography, quantum informatics, etc. One of the rapidly developing scientific and technical areas is the theory of digital antenna arrays, which has extensive applications: medical ultrasound scanning technology (on multichannel platforms with digital emitter arrays), sonar systems, seismographs, meteorological instruments, radio relay stations, avionics, radio astronomic devices, etc. The formation of these applications is accompanied by the intensive development of new computational methods.

Antenna arrays coordinately combine many individual antennas into a single system - from a few antennas to many thousands of antennas. The emergent properties of such systems provide their amazing functionality, which far exceeds the capabilities of individual antennas and causes humanity to saturate and envelop the Earth with millions of antenna arrays. Such a combined array of antennas (technical or biological) can grow and expand depending on the specific conditions of its existence, being replenished by connecting new sets of antennas to it. The spatial configuration and dimensions of antenna arrays can be very different, but they all operate on a matched emission and reception of electromagnetic and other waves by separate antennas in their composition. The chemical and structural composition of antenna arrays can also be different and include, among other things, photonic crystals and liquid crystals, examples of which are abundant in living bodies. Modern science sees great prospects with nanoantennas, which are expected to lead to revolutionary changes in computer technology (photonics) and energy (efficient use of solar energy). Nanoantennas based on DNA are already used in scientific technologies: Canadian scientists have created glowing nanoantennas from DNA molecules to track the relationships within proteins. These nanoantennas are capable of fluorescence and can absorb radiation at one wavelength and emit light at a different frequency depending on the molecular environment (Harroun et al. (2022)). This antenna is 5 nm long and is the smallest antenna ever made. It can be assumed that humanity is entering the era of the technological use of biomolecular antennas.

The importance of antenna arrays in different technical fields has led to the intensive development of the mathematical theory of transmitting and receiving antenna arrays of various types, which is presented in many publications (look at review in Petoukhov (2022a)). The mathematical description of the operation of engineering antenna arrays, operating to emit or receive waves, is almost the same. The concept of antenna arrays with its special computational methods is essential for the concept of biological computation since electromagnetic waves are capable of transmitting information in the course of biological computing, as noted in the works (Liberman (1972), Igamberdiev & Shklovskiy-Kordi (2017)). The topic of antenna arrays is important for algebraic modeling and understanding the mechanisms of genetic intelligence, which ensures the step-by-step development of a genetically encoded organism from a single fertilized cell into a single colony of trillions of mutually coordinated cells. The author's doctrine about bio-antenna arrays as the basis of energy-information biological evolution gives new approaches to study some inherited possibilities of living bodies for operations of intellectual types such as, for example, echolocations of dolphins and bats.

But in the scientific literature, it was not possible to find a single publication that would connect biological phenomena with the emergent properties of antenna arrays. Filling this gap, the author has studied and identified many heritable biological structures associated with the idea of bio-antenna arrays and their wave activity (Petoukhov (2022a)). It seems natural to assume that biological evolution has not passed by the emergent properties of antenna arrays, which have conquered the modern technology of remote interconnection and sense of the surrounding space. But in the scientific literature (before the author's publications) it was not possible to find a single publication that would connect biological phenomena with the emergent properties of antenna arrays. One can believe that tensor-unitary transformations will be useful for developing algebraic approaches in this field.

The author has studied and identified many heritable biological structures associated with the idea of bio-antenna arrays and their wave activity (Petoukhov (2022a)). It seems natural to assume that biological evolution has not passed by the emergent properties of antenna arrays, which have conquered the modern technology of remote interconnection and sense of the surrounding space. But in the scientific literature (before the author's publications) it was not possible to find a single publication that would connect biological phenomena with the emergent properties of antenna arrays. One can believe that tensor-unitary transformations will be useful for developing algebraic approaches in this field

## 3 STOCHASTIC DETERMINISM AS AN ANTIPODE TO DETERMINISTIC CHAOS

Our world is full of random events. The life and development of genetically inherited bio-bodies are inextricably linked with random interactions of molecules, against which the inherited deterministic macrostructures with the parental traits are realized. So, hidden determinism and the corresponding laws of stochastic determinism are hidden behind these accidents. The important role of probabilities in Nature is reflected in quantum mechanics based on the concept of probabilities. At the same time, as it is well-known, the Nobel laureate in physics R. Feynman noted that nobody really understands quantum mechanics. The studies of the biological dualism "stochastics-determinism" and the universal rules of the stochastic organization of genomic DNAs, presented in this paper, draw attention to the need to develop the theory of "stochastic determinism" (or "chaotic determinism") as an antipode to the well-known theory of "deterministic chaos". One can believe that in the future theory of stochastic determinism, a prominent place will be occupied by the tensor-unitary transformations described above, which model the named dualism, biological gestalt phenomena, and tensor reproduction of biostructures. Let's look at the differences between these two theories.

The theory of deterministic chaos (or dynamic chaos) has been developed by the works of a large number of mathematicians and physicists. It has been the subject of a large number of publications. Deterministic chaos is a phenomenon and a part in the theory of dynamical systems, in which the behavior of a nonlinear system looks random, despite the fact that it is determined by deterministic laws. The reason of this phenomenon is instability (sensitivity) with respect to the initial conditions and parameters: a small change in the initial condition over time leads to arbitrarily large changes in the dynamics of the system. Dynamics that is sensitive to the slightest changes in the initial conditions of the system, from which its development, change begins, and in which these slightest deviations multiply many times over time, making it difficult to predict the future states of the system, is often called chaotic. A well-known example of a system of deterministic chaos is Sinai billiards (Dorfman (1999)).

What is the difference between stochastic determinism, represented in the phenomena of biological dualism "stochastics-determinism", and deterministic chaos? Table 1 shows the main differences.

**Table 1**. The main differences between deterministic chaos and stochastic determinism

| N | Deterministic chaos | Stochastic determinism |
|---|---|---|
| 1 | The birth of the random from the non-random is observed. A completely deterministic system generates unpredictable states that have the properties of a random process. | The birth of the deterministic (non-random) from the random is observed. A system characterized by random interactions of molecules at the cellular level gives rise to deterministic biological forms. |
| 2 | A small change in the initial condition leads to arbitrarily large changes in the dynamics of the system over time (it is instability or sensitivity of the system with respect to the initial conditions). | Random interactions among molecules at the beginning of the biosystem development have little effect on the final deterministic result of its development (stability or insensitivity with respect to initial conditions). |
| 3 | In systems of deterministic chaos, it is customary to consider multi-parameter systems with a fixed dimension of their configuration spaces (for example, Sinai billiards). | In the stochastic determinism of biological systems, systems are considered with tensor development of the dimension of their configuration spaces (for example, when sets of interconnected billiards of Sinai are born tensorically in the course of development). |

One of the distinguishing features of the living is the presence of forms of constant movement in it. No wonder they say that "life is movement". Correspondingly, a living cell can always be distinguished from a dead one by this feature: in almost all biological tissues, cells move continuously, albeit slowly, or at least change shape. But it is not at all a thermal, Brownian motion. On the contrary, "*Brownian motion in a eukaryotic cell is a sign of its death*" (Fulton (1984)). Taking into account all having materials about biological dualism "stochastics-determinism", the author puts forward the following hypothesis: in living bodies, a special type of stochastic (or stochastic-deterministic) motions exist, the study of which is important for understanding biological stochastic-deterministic phenomena. It is probably that this kind of motions has its own principles of minimization and conservation laws. The algebraic toolkit described above can be useful in these future studies and in developing the theory of stochastic determinism as well. The development of the algebraic theory of stochastic determinism using tensor-unitary transformations seems useful for the formation of new approaches to genomorphic-type artificial intelligence systems based on the fundamental organization principles of genetic coded biological bodies.

## 4 CONCLUSION

The tensor-unitary transformations proposed by the author seem to be a useful tool for developing model approaches to studying genetic intelligence, coordinated growth of parts in developing multiparameter biological and technical systems, as well as the development of genetic algorithms in connection with the creation of AI. The topic of genetic intelligence is also connected with the doctrine of bio-antenna arrays as the basis of energy-information biological evolution (Petoukhov (2022a)).

## 5 ACKNOWLEDGMENTS

The author thanks Yu. I. Manin, V. I. Svirin, I. V. Stepanyan, G. K. Tolokonnikov, and S.Ya. Kotkovsky for useful discussions on the topic of tensor-unitary transformations and their applications.

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
