# OpenReview forum: "GENETIC INTELLIGENCE AND TENSOR-UNITARY TRANSFORMATIONS"
_mathai.club/MathAI/2025/Conference — MathAI 2025 Oral_

### Official Review · Reviewer_3oTq · 2025-02-24
**Review on: GENETIC INTELLIGENCE AND TENSOR-UNITARY TRANSFORMATIONS**

**Rating:** 7
**Confidence:** 4

**Review:**

Strong points:
1) The paper fits the conference "MathAI" agenda perfectly.
2) The mathematical framework proposed by the author appears promising to model the ground level of biological intelligence and hence perspective for being applied to building novel AI algorithms and systems.
3) In particular the interesting claim id posed that given framework can assure stability of emergent processes driving evolution of complex live systems capable to intelligence.

Weak points:
1) Paper is not formatted according to the conference template which MUST be fixed.
2) Paper length is less than 8 pages - MUST be fixed.
3) No experimental support is provided to validate the mathematical theory.

---

### Official Review · Reviewer_611c · 2025-02-25
**Review of "Genetic Intelligence and Tensor-Unitary Transformations"**

**Rating:** 8
**Confidence:** 3

**Review:**

The author introduces a novel algebraic framework that bridges the gap between genetic coding systems and advanced mathematical formalisms. The concept of tensor-unitary transformations, which preserve vector lengths while increasing dimensionality, is a unique contribution to the field. This approach allows for the modeling of complex biological processes, such as ontogenetic development, in a way that captures both stochastic and deterministic aspects of genetic inheritance.

Applications of tensor-unitary transformations in AI and genetic algorithms are particularly noteworthy. If successfully implemented, these ideas could lead to the development of more robust and adaptive AI systems that can evolve and grow in a manner similar to biological organisms. This could change the way we approach machine learning and AI development, making systems more resilient to noise and capable of handling complex, multiparameter environments.

This work aligns with contemporary research trends in bioinformatics and quantum computing, offering potential applications in artificial intelligence and computational genetics. However, while the theoretical novelty is evident, empirical validation of the proposed transformations is lacking. The paper does not include computational simulations or practical demonstrations that could substantiate its claims.

Strengths:
- New approach: introduction of tensor-unitary transformations provides a new and powerful tool for modeling genetic intelligence and biological development.
- Interdisciplinarity: connects bioinformatics, quantum mechanics, and artificial intelligence, fostering a multidisciplinary perspective.
- Applications in AI: the proposed applications in AI and genetic algorithms are innovative and could have significant implications for future research and technology development.

Weaknesses:
- Mathematical theory presented in the study is not supported by experimental evidence or practical validation.
- The paper does not extensively discuss the potential limitations or challenges of applying tensor-unitary transformations to real-world biological systems or AI development.
- Does not meet the conference requirement of a minimum length of 8 pages.

---

### Official Review · Reviewer_4diW · 2025-02-27
**GENETIC INTELLIGENCE AND TENSOR-UNITARY TRANSFORMATIONS**

**Rating:** 9
**Confidence:** 4

**Review:**

In the report the genetically inherited ability of living bodies to intellectual activity is considered. Genetic intelligence is understood as a part of the intellectual potential that allows on the basis of genetic information to build an organism with trillions of cells so that the parental traits are reproduced in it. It is talking about the systematic growth in the course of ontogenesis of the number of parameters and degrees of freedom of the body with a corresponding increase in the dimensionality of its configuration space of states. Genetic intelligence suppose that with such growth, the organism acquiring new degrees of freedom and knowledge, somehow retains the memory of the skills and knowledge that it possessed at previous stages of life. The author develops the algebraic foundations for modeling this fundamental feature of the development of living bodies in the tensor-matrix language of systems of multidimensional vector configuration spaces. Tensor-unitary transformations preserve the lengths of vectors during their tensor transformation into vectors of a space of increased dimension and provide expansion of stochastic-deterministic memory with preservation of all previous memory. Possible applications of tensor-unitary transformations for the development of AI, genetic algorithms, etc. are discussed.

Strengths:
1. The paper in some sense relate the conference "MathAI".
2. It is obviously original and new.
3. It have a good matter for discussion.

Weaknesses:
1. The paper need the section "Related works" that refer not only to Manin Yu. I. and other algebraic biologist, but to works from the System Biology, gene expression, systemogenesis, gene regulation and etc.
2. The paper looks as pure mathematical inverstigation it is justified. But it need to be clear settled in the section "Related works" is this work model some prossesses from System Biology, gene expression, systemogenesis, gene regulation and etc or not? And if it is then such correspondence need to be presented.
3. Paper is not formatted according to the conference template.
4. The length of the pages is less than 8.

---

### Official Review · Reviewer_ZBc1 · 2025-02-27
**Review of “GENETIC INTELLIGENCE AND TENSOR-UNITARY TRANSFORMATIONS”**

**Rating:** 7
**Confidence:** 4

**Review:**

The paper explores the concept of tensor-unitary transformations as a tool for understanding genetic intelligence and the coordinated growth of biological and technical systems. The transformations allow for the preservation and expansion of memory in multi-parameter systems, providing a novel framework for developing artificial intelligence and genetic algorithms.
Strong points:
1. The paper is in line with the theme of conference "MathAI" .
2. Provides a relatively novel mathematical framework that can provide new directions for discussion.

Weak points:
1. The paper lacks concrete experimental evidence that uses tensor-unitary transformations to support the author's conjecture.
2. The abstract and introduction are too long, and some parts seem unnecessary and can be further streamlined. For example, "the ability to echolocate in dolphins" only appears at the beginning, and there is no explanation of it's relationship with the main theme of the paper, so it is recommended to delete or add an explanation of how to use tensor-unitary transformation to model this ability later.
3. The paper does not meet the format requirements and the length of the paper main text is less than 6 pages.

---

### Decision · Program_Chairs · 2025-03-08

**Decision:**

Accept (Oral)

**Comment:**

Your article has been accepted and you can make a presentation on the article. All articles will be sorted by rating and within the available conference places one author from each article will be invited. If there are not enough places, then you will either have the opportunity to present remotely or come at your own expense!